# New Pyridinium Compound from Marine Sediment-Derived Bacterium *Bacillus licheniformis* S-1

**DOI:** 10.3390/molecules30010007

**Published:** 2024-12-24

**Authors:** Han Wang, Yifei Wang, Yanjing Li, Guilin Wang, Ting Shi, Bo Wang

**Affiliations:** College of Chemical and Biological Engineering, Shandong University of Science and Technology, Qingdao 266590, China; h15725209196@163.com (H.W.); kingsley11f115@163.com (Y.W.); 15153232290@163.com (Y.L.); 18905484623@163.com (G.W.)

**Keywords:** marine sediment-derived bacterium, *Bacillus licheniformis* S-1, pyridinium compound, cyclic dipeptides, antibacterial activity

## Abstract

The structural diversity of marine natural products is considered a potential resource for the pharmaceutical industry. In our study of marine-derived compounds, one bacterium *Bacillus licheniformis* S-1 was discovered to have the ability to produce bioactive natural products. After a further chemistry investigation, one novel 4-aminopyridinium derivative, 4-(dimethylamino)-1-(2*S*-((4hydroxybenzoyl)oxy)propyl)pyridin-1-ium (**1**), along with 15 known cyclic dipeptides (**2**–**16**) were isolated from the bacterium *B. licheniformis* S-1 derived from a shallow sea sediment. The structures of compounds **1**–**16** were elucidated through comprehensive NMR spectroscopic and specific optical rotation (OR) data analyses. Compound **6** showed antibacterial activity against *Pseudomonas fulva* with an MIC value of 50 µg/mL. This is the first study to discover a pyridinium derivative and cyclic dipeptides from *B. licheniformis.*

## 1. Introduction

The ocean covers more than three-quarters of the Earth’s surface and is rich in resources, particularly microbial resources [1]. Research has shown that secondary metabolites produced by marine-derived microorganisms possess biological activities such as antibacterial, anti-inflammatory and cytotoxic activities [2,3]. The discovery of these active compounds provides important resources for the study of marine drug lead compounds, effectively addressing the issue of drug sources in the drug development process and possessing significant medicinal value.

*Bacillus licheniformis* is widely used in industry, environmental protection, graziery and medicine. Pathak et al. isolated alkaline protease from *B. licheniformis* KBDL4, which remained highly stable under conditions of high pH and high temperature and can be applied as detergents in industry [4]. *B. licheniformis* SP34 can inhibit the growth of *Microcystis aeruginosa* DCM4 to improve water quality [5]. *B. licheniformis* is one of the probiotic strains commonly used in feed additives in graziery [6]. The addition of live *B. licheniformis* particles on the basis of conventional treatment can reduce treatment time, reduce the level of inflammatory factors and improve treatment effectiveness for children with rotavirus enteritis [7]. While there are a few studies on the secondary metabolites of *B. licheniformis*, as far as we know, only five compounds have been isolated from the bacteria until now [8,9,10].

Marine-derived *B. licheniformis* also showed attractive activity and application potential. *B. licheniformis* AS3 showed a crude oil biodegradation efficiency of 92%, indicating its potential as a biosurfactant-producing bacteria [11]. *B. licheniformis* VIT02 showed antibacterial activities against *Aeromonas hydrophila* and *Vibrio parahaemolyticus* [12]. Phelan et al. isolated three sponge-associated *B. licheniformis*, which showed good antimicrobial activities against *Bacillus cereus*, *Bacillus megaterium*, *Listeria innocua* and *Clostridium sporogenes* [13].

In our efforts to identify new bioactive secondary metabolites from marine microorganisms, we found that *Bacillus licheniformis* S-1 has the potential to produce antimicrobial secondary metabolites, and further isolation led to one new pyridinium derivative and fifteen cyclic dipeptides.

## 2. Results

Compound **1** was obtained as colorless needle crystals. Its molecular formula was determined by an HRESIMS (*m/z* [M]^+^301.14185, calculated for C_17_O_21_N_3_O_2_^+^, 301.15487) spectrum as C_17_O_21_N_3_O_2_^+^ (Appendix A), indicating 9 degrees of unsaturation. The UV spectrum exhibited absorption bands at 286 nm, indicating the emergence of conjugated structures. The NMR data (Table 1) of three methyl groups (δ_H_ 1.22 (3H, d, 6.3), δ_C_ 20.6; δ_H_ 3.24 (6H, s), δ_C_ 40.2), one methylene (δ_H_ 4.23 (1H, dd, 13.7, 2.9), 3.91–3.87 (1H, m), δ_C_ 65.0), one methine (δ_H_ 4.06–4.00 (1H, m), δ_C_ 67.5), four aromatic methines (δ_H_ 6.96 (2H, d, 7.8), δ_C_ 108.4; δ_H_ 8.09 (2H, d, 7.8), δ_C_ 143.8) and a quaternary carbon signal (δ_C_ 158.1) were almost similar to those of 4-(dimethyllamino)-1-(2*R*-hydroxypropyl)-pyridinium [14], indicating that it has a similar substructural unit to 4-(dimethyllamino)-1-(2*R*-hydroxypropyl)-pyridinium in **1**. The only difference between the substructure of **1** and 4-(dimethyllamino)-1-(2*R*-hydroxypropyl)-pyridinium was the high-field shift of C-9 (20.6 in **1** vs. 24.2 in 4-(dimethyllamino)-1-(2*R*-hydroxypropyl)-pyridinium), suggesting the different absolute structure in C-8. The other NMR data of four aromatic methines (δ_H_ 6.72 (2H, d, 8.6), δ_C_ 115.3; δ_H_ 7.82 (2H, d, 8.7), δ_C_ 132.3), two quaternary carbon signals (δ_C_ 129.8, δ_C_ 161.0) and an ester group signal (δ_C_ 175.6), combined with the ^1^H-^1^H COSY correlations (Figure 1 and Appendix A) of C-15/C-16 and C-18/C-19, the HMBC correlations from H-16/H-18 to C-14, H-15/H-19 to C-17, and H-15/H-19 to C-13, and the downfield shift of C-17 (δ_C_ 161.0), suggested the presence of a 4-hydroxybenzoyl ester substructure of **1**. The two substructures of **1** were linked through the ester group in C-13. Thus, the planar structure of **1** was determined. The substructure of compound **1** was the same as that of known compound 4-(dimethyllamino)-1-(2*R*-hydroxypropyl)-pyridinium), so the absolute configuration of **1** was further speculated to be 8*S* by comparing the specific optical rotation data with that of 4-(dimethyllamino)-1-(2*R*-hydroxypropyl)-pyridinium ($[\alpha {]}_{D}^{20}$-79.8 (c 1.0, MeOH) of **1** vs. +1100 of 4-(dimethyllamino)-1-(2*R*-hydroxypropyl)-pyridinium) [14], and it was named 4-(dimethylamino)-1-(2*S*-((4hydroxybenzoyl)oxy)propyl)pyridin-1-ium.

The structures of **2**–**16** were determined to be cyclo(D-Leu-L-Leu)(**2**) [15,16], cyclo(L-leucyl-L-leucyl)(**3**) [17,18], cyclo-(L-Pro-L-Ile)(**4**) [19,20], cyclo-(L-Pro-D-Val)(**5**) [21,22], cyclo (D-Val-D-Pro)(**6**) [23], cyclo(L-Pro-L-Val)(**7**) [24,25], cyclo (D-pro-L-val)(**8**) [26], cyclo (L-Ala-L-Pro)(**9**) [27], cyclo-(L-Pro-D-Leu)(**10**) [20,28], cyclo-(D-Pro-D-Leu)(**11**) [29], cyclo(L-Phe-L-Pro(**12**) [30], cyclo-(L-Phe-D-Pro)(**13**) [31], cyclo (L-Pro-L-Tyr)(**14**) [32], cyclo-(4methyl-D-Pro-L-Nva)(**15**) [20] and cyclo-(L-4hydroxyl-Pro-L-Leu)(**16**) [33] (Figure 2), respectively, by comparing their NMR and specific OR data (Appendix A) with those in the literature.

All compounds were tested for their inhibitory activities against fifteen pathogenic microbes. Only compound **6** showed antibacterial activity against *P. fulva* with an MIC value of 50 µg/mL.

## 3. Materials and Methods

### 3.1. General Experimental Procedure

Optical rotations were recorded using a JASCO P-1020 digital polarimeter (JASCO, Tokyo, Japan). UV spectra were measured on an Implen Gmbh NanoPhotometer N50 Touch (Implen, Munich, Germany). NMR spectra were measured using a Bruker AVANCE NEO (Bruker, Fällanden, Switzerland) at 600 MHz for ^1^H and 151 MHz for ^13^C in CDCl_3_ or DMSO. HRESIMS spectra were recorded using a Thermo Scientific LTQ Orbitrap XL spectrometer (Thermo Fisher Scientific, Bremen, Germany). HPLC separation was performed using a Waters 2989 UV/Visible detector (Waters, Milford, MA, USA) with a Kromasil 100-5-C18 HPLC column (Kromasil, Göteborg, Sweden) used at 30 °C.

### 3.2. Bacterial Material

The strain S-1 was isolated from the marine sediment collected from the Huanghai Sea in Qingdao, China, in 2021. It was identified as *Bacillus licheniformis* through the sequence analysis of the 16S rDNA internal spacer (ITS) fr (GenBank accession number PP506587). This bacterial strain is preserved at Shandong University of Science and Technology.

### 3.3. Fermentation, Extraction and Isolation

The strain *B. licheniformis* S-1 was cultured on Nutrient Broth (NB) agar plates at 37 °C for 3 d. Subsequently, it was cultivated in a NB liquid medium in 30 Erlenmeyer flasks (300 mL in each 500 mL flask) at room temperature in static. According to the actual growth state of *B. licheniformis* S-1, a 40 d period of fermentation was selected.

Each fermented culture medium was inactivated by adding 50 mL ethyl acetate (EtOAc) in static for 3 d. The mixture was filtered through two layers of gauze, and then the mycelia and the medium were separated. The mycelia and the medium were extracted three times with EtOAc, respectively. All the extracts were combined and then evaporated to dryness using a rotary evaporator to afford residue (2.46 g).

According to the polarity shown by a thin-layer chromatography (TLC) experiment, the developing agent with a retention factor value of 0.2 was selected as the mobile phase. The residue was separated into three fractions (Fr.1–Fr.3) on silica gel, with a gradient of EtOAc–PE (0–100%). Fr.1 was isolated by gel chromatography and eluted with CH_2_Cl_2_–MeOH (0–10%) to afford two fractions (Fr.1.1, Fr.1.2). Fr.1.1 was purified with HPLC eluted with 30% MeOH–H_2_O (2 mL/min) to give compounds **2** (2.3 mg, t_R_ 5.418 min), **5** (1.8 mg, t_R_ 30.230 min) and **4** (5.9 mg, t_R_ 33.725 min). Fr.1.2 was purified with HPLC eluted with 40% MeOH–H_2_O (2 mL/min) to give compounds **1** (3.3 mg, t_R_ 5.359 min) and **10** (4.6 mg, t_R_ 17.910 min). Fr.2 was isolated by gel chromatography and eluted with CH_2_Cl_2_–MeOH (0–10%) to afford three fractions (Fr.2.1, Fr.2.2 and Fr.2.3). Fr.2.2 was isolated by HPLC eluted with 60% MeOH–H_2_O (2 mL/min) to afford Fr.2.2.1 and compound **3** (2.7 mg, t_R_ 16.369 min). Fr.2.2.1 was isolated by HPLC eluted with 40% MeOH–H_2_O (2 mL/min) to afford four fractions: Fr.2.2.1.1, Fr.2.2.1.2, Fr.2.2.1.3, and Fr.2.2.1.4. Fr.2.2.1.1 was purified with HPLC eluted with 30% MeOH–H_2_O (2 mL/min) to give compounds **7** (2.7 mg, t_R_ 14.240 min) and **6** (5 mg, t_R_ 15.605 min). Fr.2.2.1.2 was purified with HPLC eluted with 30% MeOH–H_2_O to give compounds **15** (2.9 mg, t_R_ 27.515 min) and **11** (7.1 mg, t_R_ 29.976 min). Fr.2.2.1.3 was purified with HPLC eluted with 40% MeOH–H_2_O (2 mL/min) to give compound **12** (7.2 mg, t_R_ 22.618 min). Fr.3 was isolated by gel chromatography and eluted with CH_2_Cl_2_–MeOH (0–10%) to afford two fractions (Fr.3.1, Fr.3.2). Fr.3.1 was purified with HPLC eluted with 40% MeOH–H_2_O to give compounds **9** (2.1 mg, t_R_ 6.650 min), **14** (5.2 mg, t_R_ 8.406 min), **8** (3.2 mg, t_R_ 9.618 min), **16** (2.9 mg, t_R_ 11.900 min), **13** (9.5 mg, t_R_ 22.558 min) and **12** (3.2 mg, t_R_ 27.547 min).

### 3.4. Antimicrobial Activity Assays

The antimicrobial assay was evaluated by broth microdilution in 96-well plates with a sample concentration of 50 µg/mL [34]. Fifteen pathogenic microbes, namely, *Canidia albicans* (ATCC10231), *Aeromonas salmonicida* (ATCC7965D), *Escherichia coli* (ATCC 25922), *Comamonas terrigena*, *Aeromonas hydrophila* (ATCC 7966), *Photobacterium angustum* (ATCC 33975), *Photobacterium halotolerans*, *Vibrio anguillarum* (ATCC 19109), *Vibrio harveyi* (ATCC BAA-2752), *Pseudomonas aeruginosa* (ATCC 10145), *Pseudomonas aeruginosa* (ATCC 10145), *Pseudomonas fulva* (ATCC 31418), *Staphylococcus aureus* (ATCC 27154), *Enterobacter cloacae* (ATCC 39978) and *Xanthomonas axonopodis*, were used. Ciprofloxacin was used as the positive control and DMSO as the negative control.

## 4. Conclusions and Discussions

In summary, one new pyridinium derivative, 4-(dimethylamino)-1-(2*S*-((4hydroxybenzoyl)oxy)propyl)pyridin-1-ium (**1**), along with fifteen cyclic dipeptides were isolated from *Bacillus licheniformis* S-1. Compound **6** showed antibacterial activity against *P. fulva*. This is the first study to discover a pyridinium derivative and cyclic dipeptides from *Bacillus licheniformis.* The results demonstrate the significance of secondary metabolites isolated from marine-derived microorganisms in providing innovative structural templates for new drug development.

Although *B. licheniformis* is a common soil-dwelling bacteria, there are few reports of its activity from marine sources, and studies on its secondary metabolites are also limited. In the preparation experiment, the crude extract of *B. licheniformis* S-1 showed antimicrobial activity, but only compound **1** was active among the isolated compounds, which may be because the antimicrobial activity of the crude extract comprised a synergistic effect of a variety of compounds, or because the active compounds were lost during the separation process.

## Figures and Tables

**Figure 1 molecules-30-00007-f001:**
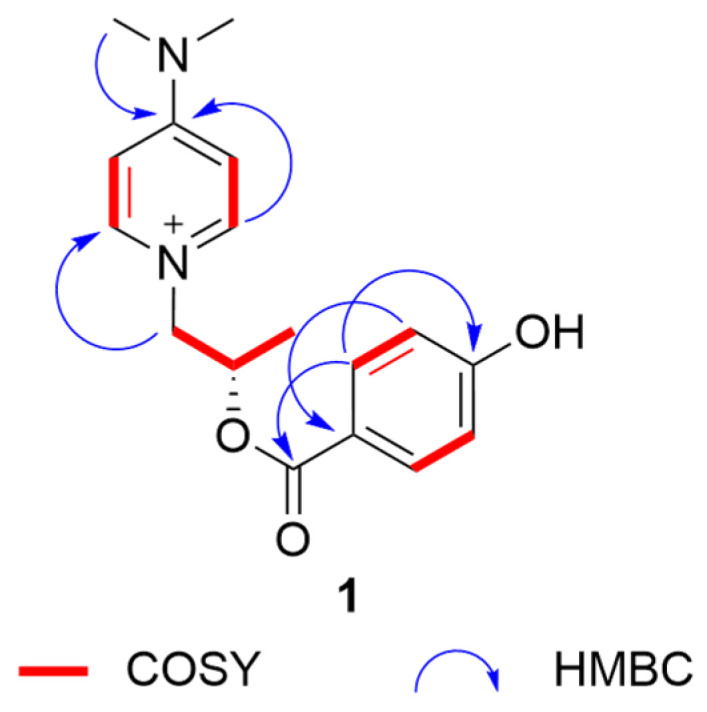
Key HMBC and COSY correlations of **1**.

**Figure 2 molecules-30-00007-f002:**
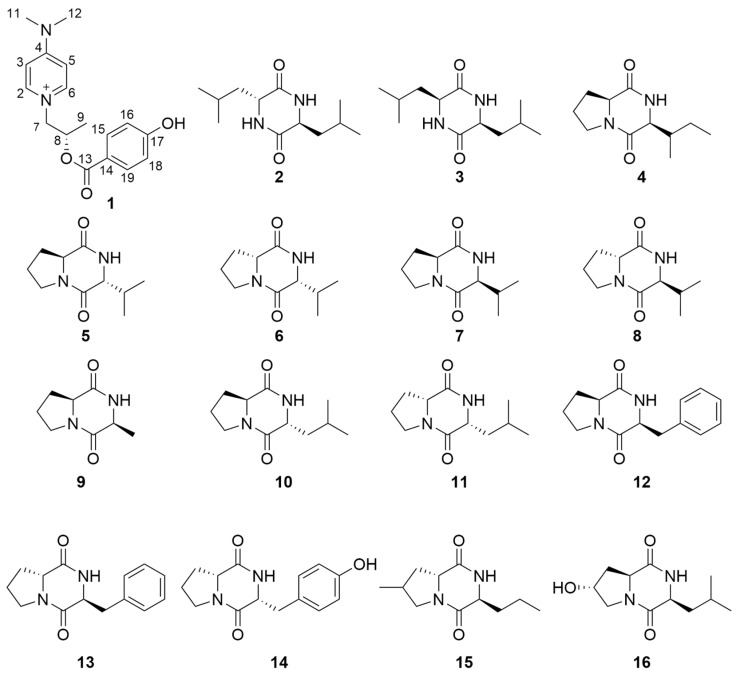
Structures of compounds **1**–**16**.

**Table 1 molecules-30-00007-t001:** The ^1^H and ^13^C NMR data of compound **1**.

	δ_C_, multi. ^a^	δ_H_ (*J*/Hz) ^a^
2, 6	143.8, CH	8.09 (d, 7.8)
3, 5	108.4, CH	6.96 (d, 7.8)
4	158.1, C	
7	65.0, CH_2_	4.23, dd (13.7, 2.9)
		3.91–3.87, m
8	67.5, CH	4.06–4.00, m
9	20.6, CH_3_	1.22 (d, 6.3)
11, 12	40.2, CH_3_	3.24, s
13	175.6, C	
14	129.8, C	
15, 19	132.3, CH	7.82 (d, 8.7)
16, 18	115.3, CH	6.72 (d, 8.6)
17	161.0, C	

^a 1^H (600 MHz) and ^13^C (151 MHz) measured in MeOD.

## Data Availability

Data are contained within the article.

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
