# Peer review of "New Pyridinium Compound from Marine Sediment-Derived Bacterium Bacillus licheniformis S-1"

_molecules, 2024, doi:10.3390/molecules30010007_

Round 1

Reviewer 1 Report

Comments and Suggestions for Authors

The manuscript “New pyridinium compound from marine sediment derived bacterium Bacillus licheniformis S-1” [molecules-3338296-peer-review-v1] written Han Wang, Yi-Fei Wang, Yan-Jing Li, Gui-Lin Wang, Ting Shi, Bo Wang describes an isolation of one new and 15 know compounds from a marine derived Bacillus licheniformis. The antibacterial activity of the compounds was also tested against 15 bacterial strains.

The reviewer has expertise in the molecular field of organic chemistry and structure determination and hence mostly refers to this part of the manuscript with the review.

All experiments and measurements are performed with modern and common state-of-the-art methods. The overall work seems relative well planned and performed. The presentation is relatively short but contains all the important details. The isolation and structure determination does (by far) not show any scientific weaknesses (see comment c). However, the results of the biotests do not show any possible targeted use. It would therefore be interesting to investigate some further bioactivities.

The results gained possess some importance in furthering our knowledge of natural products from Bacillus licheniformis isolated from unusual environment as well as of their antibacterial potential. The manuscript is of interest in the fields of Natural Product Chemistry and Phytochemistry as well as to some extent in Pharmaceutical Chemistry. It is worth publishing in general.

However, there are some weaknesses in the presentation and the manuscript hence should be revised. Therefore, the reviewer has some comments that should be considered by the authors before the manuscript can be accepted for publication in "Molecules".

General Comments:

a) In the introduction, the authors describe pharmacological effectiveness in connection with rotaviruses. However, the selected biotests deal with activity against other bacteria and do not show great effectiveness. The reviewer therefore recommends that the authors conduct additional biotests with regard to activity against rotaviruses. Or with regard to inhibition/activation of enzymes that may be involved in such context. If some activity is shown here, this data would further increase the quality of the manuscript, as a possible use would be demonstrated.

b) Bacillus licheniformis is a microorganism found ubiquitously in soil. Authors are encouraged to write at least something about the unusual(?) marine source of the bacterium and thereby clarify the importance of its isolation from this marine environment.

c) The reviewer is not entirely clear how the authors determined the absolute configuration from the optical rotation value. The comparison molecule is relatively different in structure, so that a definite conclusion about the stereochemistry does not seem to be 100% sure. The authors are therefore encouraged to investigate this in more detail; if necessary with a CD spectrum and a corresponding calculation/simulation, from which the absolute configuration could be derived more reliably.

d)
A graphical representation of the separation process (in the Supp. Mat.) would be helpful in understanding.

e) All authors should be listed in all references!

Minor Comments:

f) Line 4: Gui-Lin Wang does not have an affiliation indexed. Please add.

g) Line 16: Write out "Pseudomonas fulva" here.

h) Line 19: Write “Bacillus licheniformis” in italics.

i) Line 34 and elsewhere: There should be a space between the word and the reference: "... quality [5]."

j) Line 75 and elsewhere: The stereo indicators of the Fischer nomenclature ("D" and "L") are to be set in small capitals. Authors are encouraged to do so throughout the manuscript.

k) Lines 92-94 This/These sentence(s) is/are not clear to the reviewer.

l) Line 104 (and elsewhere): Use the abbreviations of the SI units (“d” instead of “days”).

m) Line 132: Pseudomonas aeruginosa with “P” in italics.

n) In some references the page number are missing. (Ref. 1, 2, 11, 17, 22) Please add them all.

o) Lines195-197: In reference 20 there is somehow a typesetting mistake.

p) Lines 219-221: Please remove this information.

q) In the Supplementary the first headline is in Chinese. It might be confusing for readers to switch between languages.

Author Response

Thank you very much for your careful reading and valuable comments. After receiving your feedback, we have made changes accordingly and invite you to review them again. If the article needs further improvement or has any questions, we would be grateful for your continued guidance.

General Comments:

a)In the introduction, the authors describe pharmacological effectiveness in connection with rotaviruses. However, the selected biotests deal with activity against other bacteria and do not show great effectiveness. The reviewer therefore recommends that the authors conduct additional biotests with regard to activity against rotaviruses. Or with regard to inhibition/activation of enzymes that may be involved in such context. If some activity is shown here, this data would further increase the quality of the manuscript, as a possible use would be demonstrated.

Response: Thank you for your kind remind. The pharmacological effectiveness in connection with rotaviruses described in the introduction was used live B. licheniformis particles, which are the action of live bacteria rather than secondary metabolites. We found that the crude extract had antimicrobial activities, so microorganisms were selected for our subsequent activity test. We added tables S2 and S3 in the supplementary materials, which contain the antimicrobial activities of all compounds, and added the discussion section, which included the reasons why the active compounds were not distributed in lines 162-165.

b)Bacillus licheniformis is a microorganism found ubiquitously in soil. Authors are encouraged to write at least something about the unusual(?) marine source of the bacterium and thereby clarify the importance of its isolation from this marine environment.

Response: Thank you for your kind remind. However, after a literature research, we did not find significant difference between Bacillus licheniformis of marine and soil origin. We supplement the activity of marine sources Bacillus licheniformis in the third paragraph of the introduction and lines 160-161.

c)The reviewer is not entirely clear how the authors determined the absolute configuration from the optical rotation value. The comparison molecule is relatively different in structure, so that a definite conclusion about the stereochemistry does not seem to be 100% sure. The authors are therefore encouraged to investigate this in more detail; if necessary with a CD spectrum and a corresponding calculation/simulation, from which the absolute configuration could be derived more reliably.

Response: Thank you for your kind remind. Because of the using of bioactivity evaluation and other experiments, the quantity of the new compound remained was insufficient to support CD spectroscopy experiment. The new compound has only one chiral carbon to determine the optical rotation, The substructure of compound 1 is the same as that of known compound 4-(dimethyllamino)-1-(2R-hydroxypropyl)-pyridinium), so the absolute configuration can be speculated by the optical rotation. We have corrected this part in lines 71-73.

d)A graphical representation of the separation process (in the Supp. Mat.) would be helpful in understanding.

Response: Thank you for your kind remind. We made a graphical representation of the separation process and put it in the supplementary material in Figure S51.

e)All authors should be listed in all references!

Response: Thank you for your kind remind. We completed all authors in the references.

Minor Comments:

f)Line 4: Gui-Lin Wang does not have an affiliation indexed. Please add.

Response: Thank you for your kind remind. We have added the affiliation indexed of Gui-Lin Wang in line 4.

g)Line 16: Write out "Pseudomonas fulva" here.

Response: Thank you for your kind remind. We have written "Pseudomonas fulva" in line 16.

h)Line 19: Write “Bacillus licheniformis” in italics.

Response: Thank you for your kind remind. We have written “Bacillus licheniformis” in italics in line 19.

i)Line 34 and elsewhere: There should be a space between the word and the reference: "... quality [5]."

Response: Thank you for your kind remind. We checked all the references and added spaces where they were missing.

j)Line 75 and elsewhere: The stereo indicators of the Fischer nomenclature ("D" and "L") are to be set in small capitals. Authors are encouraged to do so throughout the manuscript.

Response: Thank you for your kind remind. We have written the stereo indicators of the Fischer nomenclature ("D" and "L") in small capitals in the manuscript from lines 84-89.

k)Lines 92-94 This/These sentence(s) is/are not clear to the reviewer.

Response: Thank you for your kind remind. We checked the sentence and found an extra “.”, which may have been entered accidentally and has now been removed in line 101-103.

l)Line 104 (and elsewhere): Use the abbreviations of the SI units (“d” instead of “days”).

Response: Thank you for your kind remind. We use “d” instead of “days” in lines 112, 114 and 116.

m)Line 132: Pseudomonas aeruginosa with “P” in italics.

Response: Thank you for your kind remind. We italicized P in line 148.

n)In some references the page number are missing. (Ref. 1, 2, 11, 17, 22) Please add them all.

Response: Thank you for your kind remind. We supplemented the page numbers in the references, but some journals did not provide the page numbers.

o)Lines195-197: In reference 20 there is somehow a typesetting mistake.

Response: Thank you for your kind remind. We fixed the mistake here and checked to make sure there are no typographical mistake now.

p)Lines 219-221: Please remove this information.

Response: Thank you for your kind remind. We deleted this paragraph

q)In the Supplementary the first headline is in Chinese. It might be confusing for readers to switch between languages.

Response: Thank you for your kind remind. We have changed the first headline in the supplementary to English.

Reviewer 2 Report

Comments and Suggestions for Authors

1)                  In the Abstract line 16 defines OR.

2)                   The results section lacks a discussion of the UV-visible spectra of the compounds obtained, which should be added to the manuscript.

3)                  The compounds should be characterized by other spectroscopic techniques such as infrared spectroscopy.

4)                  The compounds obtained should be characterized in the solid state.

5)                  No results on the HPLC separation of the compounds have been included in the results section. The experimental section should include the conditions of separation of the components of the mixture, data on the column used, flow rate, detection, temperature, etc.

6)                  In the experimental section line 105 defines EtOA and NB liquid medium in line 103.

7)                  The results obtained from the microbiological assays should be included in the manuscript, as the final MIC value of only one of the isolated compounds and a discussion related to the results obtained must be added.

8)                  In the conclusion section the sentence “The results indicate that this bacterium has the potential to produce novel and bioactive compounds” is not supported by the results obtained since only one of the isolated compounds presented antimicrobiological activity in one of the evaluated strains.

Comments on the Quality of English Language

 English needs to be improved.

Author Response

1) In the Abstract line 16 defines OR.

Response: Thank you for your kind remind. We define OR in line 16.

2) The results section lacks a discussion of the UV-visible spectra of the compounds obtained, which should be added to the manuscript.

Response: Thank you for your kind remind. We discussed the UV-visible spectra of compound 1 in lines 54-55.

3) The compounds should be characterized by other spectroscopic techniques such as infrared spectroscopy.

Response: Thank you for your kind remind. Because of the using of bioactivity evaluation and other experiments, the quantity of the new compound remained was not enough for infrared spectroscopy.

4) The compounds obtained should be characterized in the solid state.

Response: Thank you for your kind remind. Because of the using of bioactivity evaluation and other experiments, the quantity of the new compound remained was insufficient to support solid-state characterization.

5) No results on the HPLC separation of the compounds have been included in the results section. The experimental section should include the conditions of separation of the components of the mixture, data on the column used, flow rate, detection, temperature, etc.

Response: Thank you for your kind remind. For the convenience of reading, we put the HPLC separation results in the third paragraph of 4.3. We have supplemented this information, now in lines 101-103 and 124-141.

6) In the experimental section line 105 defines EtOA and NB liquid medium in line 103.

Response: Thank you for your kind remind. We defined EtOAc and NB in lines 111 and 115, respectively, and supplemented “Table S4. Acronym list”

7) The results obtained from the microbiological assays should be included in the manuscript, as the final MIC value of only one of the isolated compounds and a discussion related to the results obtained must be added.

Response: Thank you for your kind remind. We added the results of all compound microbiological assays to the supplement material (Tables S2-S3).

8) In the conclusion section the sentence “The results indicate that this bacterium has the potential to produce novel and bioactive compounds” is not supported by the results obtained since only one of the isolated compounds presented antimicrobiological activity in one of the evaluated strains.

Response: Thank you for your kind remind. The sentence “The results indicate that this bacterium has the potential to produce novel and bioactive compounds” has been deleted, and we added a discussion section for this question in lines 92-96.

Reviewer 3 Report

Comments and Suggestions for Authors

After reading the article New pyridinium compound from marine sediment derived bacterium Bacillus licheniformis S-1, the results presented by the authors are interesting, but the authors have to work on several aspects, since there is missing information.

1. Please attach the results obtained from the antimicrobial activity of compound 6, since that information is missing.

2. The discussion section is missing.

3. Attach an experimental dosage section to the material and methods section.

4. What concentrations did you use to evaluate the antimicrobial activity?

5. Add the doi to the references.

After making the suggestions, you can submit it again.

Author Response

Thank you very much for your careful reading and valuable comments. After receiving your feedback, we have made changes accordingly and invite you to review them again.

  1. Please attach the results obtained from the antimicrobial activity of compound 6, since that information is missing.

Response: Thank you for your kind remind. We added the results of all compound antimicrobial assays to the supplement material (Tables S2-S3).

  1. The discussion section is missing.

Response: Thank you for your kind remind. We added a discussion section in lines 160-165.

  1. Attach an experimental dosage section to the material and methods section.

Response: Thank you for your kind remind. We supplement the experimental dosage in line 119.

  1. What concentrations did you use to evaluate the antimicrobial activity?

Response: Thank you for your kind remind. We supplement the concentrations of the antibacterial activity experiment in line 144.

  1. Add the doi to the references.

Response: Thank you for your kind remind. We add the doi to the references.

Reviewer 4 Report

Comments and Suggestions for Authors

I propose the following improvements:

- the paper uses a lot of acronyms in a limited space, which makes it difficult to read and I think a list of the acronyms used would be welcome

- section 2.3. Fermentation, Extraction and Isolation requires an extension with additional explanations related to the path followed (e.g.: "Subsequently, it was cultivated in NB liquid medium in 30 Erlenmeyer 103 flasks (300 mL in each 500 mL flask) at room temperature in static for 40 days" - why 40 days, what is the reference, etc.)

- also in the same section, I believe that the preparation of samples for analysis requires more detailed explanations (e.g. why certain concentrations were chosen, comparisons with similar approaches presented in the literature)

Author Response

- the paper uses a lot of acronyms in a limited space, which makes it difficult to read and I think a list of the acronyms used would be welcome

Response: Thank you for your kind remind. We added “Table S4. Acronym list” in the supplementary materials

- section 2.3. Fermentation, Extraction and Isolation requires an extension with additional explanations related to the path followed (e.g.: "Subsequently, it was cultivated in NB liquid medium in 30 Erlenmeyer 103 flasks (300 mL in each 500 mL flask) at room temperature in static for 40 days" - why 40 days, what is the reference, etc.)

Response: Thank you for your kind remind. We selected fermentation for 40 days according to the actual growth state of bacteria and supplement it in lines 113 and 114. We explained the subsequent separations including selection of developing agent and flow rate and retention time of liquid phase, etc in lines 120-141.

- also in the same section, I believe that the preparation of samples for analysis requires more detailed explanations (e.g. why certain concentrations were chosen, comparisons with similar approaches presented in the literature)

Response: Thank you for your kind remind. Before selecting the sample preparation concentration, we performed thin layer chromatography (TLC) pre-test and selected the elution concentration according to the polarity of the substance, and supplemented it in the lines 120-121.

Round 2

Reviewer 1 Report

Comments and Suggestions for Authors

The manuscript entitled “New pyridinium compound from marine sediment derived bacterium Bacillus licheniformis S-1” [molecules-3338296-peer-review-v2] submitted by Han Wang, Yi-Fei Wang, Yan-Jing Li, Gui-Lin Wang, Ting Shi, Bo Wang has been revised by the authors. The reviewer is grateful to the authors for taking to heart the previous comments, also from other reviewers and for addressing several concerns.  

A very detailed list with the answers of the authors to all reviewers’ comments, point by point, is added to the new submission. The authors have taken all of the comments of the reviewer as well as obviously of other reviewers into account. The authors have answered to the concerns and made some comprehensive respective corrections, additions, and changes in the newly submitted manuscript. All changes in the manuscript make sense. They have been carried out with direct reference to the comments.

Hence, the quality and readability of the manuscript have been improved in the newly submitted version. The manuscript now fully describes an isolation of one new and 15 know compounds from a marine derived Bacillus licheniformis as well the antibacterial activity of the compounds against 15 bacterial strains. From the reviewer's perspective this manuscript is therefore acceptable for publication in the “Molecules”.

Reviewer 2 Report

Comments and Suggestions for Authors

The authors have made satisfactory changes to the manuscript as suggested by the reviewer

Reviewer 3 Report

Comments and Suggestions for Authors

The authors made the suggestions.